# Gut Microbiome and Neurodevelopmental Disorders: A Link Yet to Be Disclosed

**DOI:** 10.3390/microorganisms11020487

**Published:** 2023-02-15

**Authors:** Zoi Iliodromiti, Anastasia-Rafaella Triantafyllou, Marina Tsaousi, Abraham Pouliakis, Chrysa Petropoulou, Rozeta Sokou, Paraskevi Volaki, Theodora Boutsikou, Nicoletta Iacovidou

**Affiliations:** 1Neonatal Department, Medical School, Aretaieio Hospital, National and Kapodistrian University of Athens, 15772 Athens, Greece; 22nd Department of Pathology, “ATTIKON” University Hospital, National and Kapodistrian University of Athens, 12461 Athens, Greece; 3NICU, Nikea General Hospital “Agios Panteleimon”, 18454 Piraeus, Greece

**Keywords:** children’s health, gut microbiome, breastfeeding, neonates, neurodevelopmental disorders

## Abstract

Τhe importance of the gut microbiome and its functions has only recently been recognized and researched in greater depth. The establishment of the human gut microbiome begins in utero, forming its adult-like phenotype in the first 2–3 years of life. Several factors affect and alter the gut microbiome composition and its metabolic functions, such as early onset of breastfeeding, mode of delivery, antibiotic administration, or exposure to chemical substances, among others. Existing data support the important connection between health status and gut microbiome homeostasis. In cases when this balance is disturbed, several disorders may arise, such as inflammatory reactions that lead to atopy, eczema, or allergic asthma. The so-called gut-brain axis refers to the complex biochemical pathways between the central nervous system and the gastrointestinal system. One of the most fascinating areas of ongoing research is the broad spectrum of neurodevelopmental disorders (NDDs) and how gut health may be associated with such disorders. The prevalence of NDDs, such as autism spectrum disorder or attention deficit hyperactivity disorder, has increased over recent years. Whether gut microbiota homeostasis plays a role in these disorders is not yet fully understood. The aim of this narrative review is to provide an account of current knowledge on how gut health is linked with these disorders. We performed a literature review in order to identify and synthesize available data that highlights the potential association between NDDs and a balanced gut microbiome in terms of composition and proper function. The connection between the gut microbiome and NDDs offers promising new opportunities for future research.

## 1. Introduction

The World Health Organization defines health as “the state of complete physical, mental and social well-being and not merely the absence of disease or infirmity” [1]. However, children’s health cannot achieved optimal status on its own. Factors such as maternal health, lifestyle, health status during pregnancy, family background, and education seem to play an important role in shaping a young individual’s health status [1]. Good health during childhood will lay the foundation for decent health status as an adult, and will reduce the likelihood of chronic health problems, such as obesity, hypertension, or diabetes [2]. Recent studies have posed the hypothesis that the developmental origin of health can be traced back to as early as intrauterine life. The identification of genetic and epigenetic mechanisms that contribute to shared molecular pathways between early life and adulthood health issues, such as fetal growth and chronic diseases, is the focus of the current study [3].

One of the earliest health parameters, which is a reflection of the host’s lifestyle and behavior patterns, is the human gut microbiome. The development of intestinal microbiota begins in utero, although it is mostly established after birth [4]. Studies reveal the significance of many factors that contribute to the gut microbiota composition and function, such as mode of delivery, type of feeding (i.e., breast vs. exclusively formula feeding), antibiotic administration, duration of hospitalization, etc. [5,6]. Most of the microbiome’s development in the earliest stages of human life as well as the major changes in its composition are detected in the first 2–3 years of life [2]. A complete understanding of the gut microbiota function is still lacking, and its potential contribution to all aspects of health and disease status remains a scientific challenge.

One of the most intriguing aspects of current research is the broad spectrum of neurodevelopmental disorders (NDDs), a very wide group of disabilities primarily associated with the growth and functioning of the human brain and/or the central nervous system (CNS). A thorough understanding of their pathophysiology is still lacking, and their presentation varies [7]. Existing data provide evidence of gut dysbiosis or gut microbiota with unique patterns in hosts diagnosed with NDDs. Moreover, gastrointestinal comorbidities are frequently reported [8]. Whether there is an association between the earliest stages of gut microbiome establishment and the possible development of NDDs continues to be poorly understood. For this purpose, we conducted a narrative review of the current literature published in the English, Greek, or German languages. We searched online database platforms and retrieved papers on the association between neurodevelopmental disorders and the gut microbiome. We also examined the citations of the retrieved scientific papers. Finally, data contributing to the objective of this review was extracted from the selected papers.

The aim of this literature review is to gather existing knowledge on the topic of NDDs and their association with the gut microbiome to establish new scientific hypotheses for future research.

## 2. Gut Microbiome

Gut microbiota refers to the trillions of microorganisms that reside in the human gut. These include bacteria, viruses, archaea, eukaryotes, and others that are present in the environment of the human intestine [9]. The term “microbiota” refers to the microbes themselves, while the term gut “microbiome” refers to the collection of microorganisms, their genomes, and their products that colonize the human intestine [9,10]. The gut is mostly colonized by phyla such as *Bacteroides, Proteobacteria, Actinobacteria*, and others, which are the predominant strains. When it comes to genera, the most common are *Streptococcus*, *Pseudomonas*, *Bacteroides*, *Fusobacteria*, *Clostridium*, and *Lactobacillus*, among others [10].

### 2.1. Functions of the Gut Microbiome

The complexity of the intestinal microbiome’s function has only recently been understood. Its key role is the direct inhibition of pathogens [10]. The host gut microorganisms consume the available nutrients and produce cytokines, which kill or inhibit hostile microbes [11], maintain the structural integrity of the mucosal barrier [12], and contribute to the nutrient metabolism of the host. They are also involved in drug uptake [13], vitamin synthesis, and the production of short-chain fatty acids as well as some gut hormones [13]. In humans, the influence of the gut microbiome is not limited to the metabolic pathways and immune system [14]; disruption of the gut microbiota’s function could lead to imbalances, giving rise to an inflammatory state and inflammatory-related diseases [14]. It has also been reported that depression and anxiety disorders are associated with an inflammatory state [15]. This phenomenon is attributed to the existence of the gut-brain axis, which is discussed in another section of this review.

### 2.2. Establishment of the Gut Microbiome

The establishment of the gut microbiome is of crucial importance as it regulates a plethora of functions in the human body. The dogma of a “sterile intrauterine environment” and the theory of the “beginning of microbial colonization at birth” have recently been challenged. Studies report that fetuses are exposed to their first inoculum of microbes in utero and the earliest stage of gut colonization begins within the amniotic cavity [4,16]. Various bacterial species have been found in samples from the placenta, the amniotic fluid, and the umbilical cord of full-term healthy neonates [17,18]. The genera of *Bifidobacterium* and *Lactobacillus* were present in placental biopsies after elective cesarian section (CS) at term [19]. At and after birth, the maturation and development of gut microbiota is a dynamic process that is affected by several factors, such as mode of delivery and gestational age; postnatal factors, such as the type of feeding, also play a role.

### 2.3. Important Factors That Influence Gut Microbiota

Figure 1 depicts the factors that contribute to the establishment of the gut microbiota. These factors are discussed in greater detail in the following paragraphs.

#### 2.3.1. Mode of Delivery

The microbiota of neonates born vaginally differ remarkably from that of neonates delivered by CS. Neonates born vaginally are directly exposed to maternal vaginal and fecal microbiota, hence they are colonized by microorganisms such as *Lactobacillus* [6,20]. Dominguez et al. report that vaginally born neonates acquired bacterial communities that resembled both the vaginal and intestinal microbiota of their mothers. The dominant species were *Prevotella*, *Lactobacillus*, and others [6]. The same study found that skin surface cultures of neonates delivered by CS were mostly dominated by *Staphylococcus*, *Propionibacterium*, and *Corynebacterium*, species that are found on the skin [14], and by microorganisms from the hospital environment [21,22]. In addition, fecal microorganisms of infants born by vaginal delivery resemble the gut microbiota of their mothers by the third day of life, which is not observed in babies delivered by CS [23]. The observed differences associated with the delivery mode are possibly linked with the protective nature of vaginal birth when it comes to long-term indices [6]. A newborn baby’s microbiota starts to develop during the first days of life; however, this microbiota lacks the profound diversity of commensal bacteria, which are established later.

#### 2.3.2. Gestational Age

Depending on the week of gestation, prematurity has been linked with serious health challenges. Preterm neonates are born with immature systems, such as immune, respiratory, gastrointestinal, and neurological [24]. They sometimes need prolonged hospitalization as they may require, among other interventions, mechanical ventilation or the intravenous administration of drugs such as broad spectrum antibiotics; they may also need parenteral nutrition for a prolonged period of time [24]. All these factors play a significant role in the acquisition and development of abnormal gut microbiota. Studies show that prematurity is associated with a delay in the gut’s colonization with bacteria such as *Bacteroides* or *Bifidobacterium*, species found in feces [25,26]. Aujoulat et al. report that the gut microbiota of preterm infants is mostly dominated by *Staphylococcus*, *Enterococcus*, and other gram-positive microorganisms, whereas gram-negative bacteria, such as *Enterobacteriaceae*, are present in greater variability [24,27]. The gut microbiota of very low birth weight neonates seems to evolve with time, showing a gut well colonized by anaerobes, mostly *Clostridia*, which suggests a delay in the transition to the adult-type microbiome [28]. Other studies provide evidence of the correlation between the gut microbiome of preterm infants and the increased risk of sepsis and necrotizing enterocolitis [29,30]. The different composition of intestinal microorganisms seems to also play a role in microbiome functionality. Arboleya et al. report that preterm neonates produce lower levels of short-chain fatty acids when compared with full-term neonates [31].

#### 2.3.3. Feeding Mode

Feeding mode is a factor contributing to gut microbiota that has been studied extensively [6]. Microbial composition of the gut seems to differ significantly between exclusively breastfed and formula-fed infants. A major difference is that the gut microbiota of breastfed babies is less diverse. This means that it has a lower alpha-diversity and it mostly consists of milk-oriented microorganisms such as *Bifidobacteria* [32,33,34]. In particular, the infant gut is mainly colonized by the species, *Bifidobacterium longum* and *Bifidobacterium breve*. Human milk oligosaccharides (HMOs) are the third most abundant component in human milk, and while these complex saccharides cannot be digested by the infant, they act as natural prebiotics for the propagation of the beneficial *Bifidobacteria* [34,35,36]. Schwarz et al. report that exclusively breastfed infants have a higher expression of microbial virulence factors, with the simultaneous downregulation of inflammatory genes. This could highlight the possible protective role of the gut microbiome in breastfed infants [37]. In contrast, the intestinal microbiome of formula-fed infants is characterized by diversity, and consists of cocci such as *Staphylococcus* and *Enterococcus*, *Bacteroides*, and *clostridia* [20]. Bäckhed et al. report that formula feeding leads to early divergence and maturation to an adult-like microbiota [33]. The current knowledge is derived from studies on exclusively breastfed vs. formula-fed infants, which indicates the need for further research on how mixed feeding could potentially influence the gut microbiome of infants. The cessation of breastfeeding could increase the abundance of *Bacteroides*, among others, and decrease the abundance of *Bifidobacterium* [38].

#### 2.3.4. Environmental Factors (Geographic Location, Family Home, and Others)

Geographic location and family environment may be factors that influence the gut microbiota. The difference between a rural and an urban setting seems to be related to different dietary patterns. Existing data confirm that ethnicity may be related to a specific diet, and hence, a unique gut microorganism composition [39]. It is hypothesized that this difference in gut microbiota composition could be due to geographic factors, and shows a geographic gradient [39]. When fecal samples from infants living in different countries were compared, it was and observed that children living in Bangladesh had distinct gut microbiota compared with those living in a suburban area in the USA [40]. Another aspect that is being researched is the “sibling effect,” which supports the theory that infants with older siblings have an enriched, more diverse gut microbiota composition. Penders et al. report the existence of greater *Bifidobacteria* concentration in children with older siblings than in those without older siblings [20]. A Danish cohort confirmed the former theory, and also included the factor of household pets, although no significant findings were reported for this parameter [41]. However, this study showed no significant association between gut microbiota characteristics and the occurrence of the studied inflammatory diseases (asthmatic bronchitis, eczema). The effect of environmental factors (family members, house setting, ethnicity) remains controversial and requires further investigation.

#### 2.3.5. Host Genetics

An increased number of studies provides evidence on the contribution of the human host genotype to the establishment, propagation, and diversity of the gut microbiome [42,43]. However, current data do not provide clear proof to support this idea, and further research is necessary.

### 2.4. Epigenetic Regulation

The balance of the relationship between the host and the trillions of commensal gut microbes is extremely sensitive. The intestinal microbiota not only has a strong impact on host health via chemical pathways and signals, but it can also alter host cell responses through modifications to the host epigenome. These epigenetic functions include DNA methylation, regulation of non-coding RNAs, and histone modification [44]. Such epigenetic mechanisms were identified in local intestinal and peripheral cells [44]. These modifications can be adjusted by metabolites generated by the gut microbiota, such as short-chain fatty acids, biotin, trimethylamine-N-oxide, and folate [45]. In certain diseases, such as diabetes mellitus, the epigenetic connection between the gut microbiome and the host has been extensively investigated [46]. However, further prospective studies are required on neonatal populations (high risk populations, such as preterm infants and infants of diabetic mothers) and their respective gut microbiome composition to gain greater insights into this complex relationship.

Overall, the combination of these aforementioned factors seem to account for the unique gut microbiota composition of each human host (Figure 1).

### 2.5. Gut Microbiome and Association with Health-Related Issues

Changes in human gut microbiota may be a possible risk factor for a range of health-related problems, such as metabolic (i.e., diabetes mellitus) and atopic disorders [47]. The possible association between atopy and gut microbiota was first identified by Holt and Bjorksten [47,48]. Bisggard et al. report that reduced diversity of the intestinal human gut microbiota of infants aged 1 and 12 months is associated with increased risk of allergic disease during school age [49].

### 2.6. Gut Virome/Mycome: Composition and Establishment

The gastrointestinal tract is normally the location of the most abundant viral colonization, with an estimated ~109 Virus-like particles (VLPs) per gram of intestinal content [50]. Electron microscopy of the stool indicates that most of the phages belong to the *Caudovirales* order [51]. Metagenomic sequencing of the human gut virome reveals that *Caudovirales* is normally predominant, along with *Microviridae* [51,52]. Healthy term neonates at birth lack a gut virome, which matures with progressive age [50,53], and frequently undergoes a shift from being *Caudovirales*-dominated to being *Microviridae*-dominated [50].

Although various studies have been published on the gut microbiota composition and its function, these are primarily focused on bacteria. Consequently, the term “gut mycome” is still unclear and not fully studied [54].

## 3. Neurodevelopmental Disorders

The gut microbiome has a remarkable impact on the physiology of the human nervous system. From the very early stages of fetal development, the initial establishment of the microbiome is synchronized with the development of the nervous system. Promising studies suggest the actual involvement of the microbiome in regulating brain development. Nevertheless, any disturbance that could negatively influence brain functionality could also possibly lead to a range of neurodevelopment disorders [14,55,56,57].

### 3.1. Terminology and Most Recent Classification According to DSM-5

The term “neurodevelopmental disorders” is not only extremely broad, but is also sometimes difficult to conceptualize, even by physicians [7]. The most recent edition of the Diagnostic and Statistical Manual of Mental Disorders (Fifth Edition, DSM-5) classifies NDDs as autism spectrum disorder (ASD); attention deficit hyperactivity disorder (ADHD); intellectual development disorder (ID); specific learning disabilities (i.e., dyscalculia or dyslexia); communication, speech, or language disorders; conduct disorders; motor disorders (e.g., tic disorders, such as Tourette’s syndrome); cerebral palsy (which falls into the category of congenital traumatic brain injury); and Fetal Alcohol Spectrum Disorder (FASD) [7]. Neurogenetic disorders, such as Rett’s Syndrome, Down’s Syndrome (Trisomy 21), or Fragile X Syndrome, have been added to the expanded list of neurodevelopmental disabilities [7,58].

### 3.2. NDDs and Early Manifestation (Prenatal, Neonatal, Infant, Early School Age)

The characteristic neurodevelopmental deficits present in such disorders affect and impair everyday life at multiple levels [7]. People diagnosed with NDDs have impaired functioning in personal, academic, professional, and social aspects of life [59]. Brain functions, such as emotion, self-control, learning ability, memory, intelligence, and social skills, are some of the affected brain activities [7,59]. The onset of the initial symptoms and behavioral disabilities usually occurs early in childhood, but the full spectrum of NDDs generally unfolds as the individual develops and grows [59]. The range of developmental deficits tends to last over the individual’s lifetime [59].

### 3.3. NDDs and Quality of Life

The World Health Organization defines quality of life (QoL) as “a person’s perception of their position in life,” and includes an individual’s physical health, personal beliefs, psychological status, and social interactions, among others [60]. Assessment of QoL may reflect a more holistic view of the impact of health problems [61]. Children may begin reporting their own status of health-related quality of life (HRQL) as early as 4 to 6 years of age. The patient’s age may affect the type of expected symptoms, such as in ADHD. Studies on children with ADHD identify significantly lower self-reported QoL than studies on children with a long-term physical disease (i.e., diabetes mellitus) [61,62]. However, Varni et al. report that the QoL of children with ADHD does not differ significantly between children with cerebral palsy and cancer [63]. Other studies report that patients with ASD also had significantly poorer self-reported QoL [62,64]. No difference was found when the QoL reported by ASD patients with intellectual disability was compared with that reported by patients with ADHD [65]. Further research is needed on the wide spectrum of NDDs for both patients and parents or carers.

## 4. The Gut-Brain Axis

The gut-brain axis is a bidirectional biochemical signaling route between the gastrointestinal tract and the CNS. This term has recently been extended to “microbiota-gut-brain axis” or MGB axis, indicating the role of gut microbiota in the biochemical events occurring between these two systems [66]. The axis includes a plethora of systems, such as the neuroendocrine system, the neuroimmune system, the autonomic nervous system, the hypothalamic-pituitary-adrenal axis (HPA axis), the enteric nervous system, the vagus nerve, and the microorganisms of the human intestine [5,67]. Starting from birth, the chemicals that are released by the gut microbiota exert a significant influence on brain development [68]. It seems that the bidirectional communication of the axis is mediated by endocrine, neural, humoral, and immune networks [68]. A variety of chemical substances, including chemokines, cytokines, neurotransmitters, microbial metabolites, and neuropeptides, are part of the signaling [69]. The gut microbiota further relocates these chemicals to the brain via the bloodstream, the neurons, or the cells of the endocrine system [70]. When the chemicals reach specific brain locations, affect impact metabolic procedures. Irritable bowel syndrome (IBS) is a disorder of the interaction between the brain and the gut, and is directly affected by the gut microbiome [69,70]. The role of the gut-brain axis is well confirmed in this disorder. However, for many other disorders, the mechanisms need to be further studied and understood. Clapp et al. suggest that mental health and mental disorders are strongly influenced by gut microbiota. They propose that probiotics could potentially restore the normal microorganism balance in the human intestine; therefore, they may play a role in therapy and the prevention of mental issues, such as stress and anxiety disorders [71]. Wang et al. studied the positive probiotic-prebiotic effects in children diagnosed with ASD and reported reduced levels of pathogenic bacteria (*Clostridium*) and elevated levels of beneficial ones (i.e., *Bifidobacteriales*). The severity of ASD and gastrointestinal symptoms was also diminished [72]. The composition of the gut microbiome is complex, and a more holistic approach is required to understand the gut-brain axis and the treatment of these disorders.

### The Gut-Brain Axis: Function of the Gut Microbiome in Brain Development and Pathways

A growing number of studies concentrates on the interaction between the gut microbiome and the CNS. These include neurogenesis, myelination, maturation of microglial cells, development and preservation of blood-brain barrier integrity, development of the HPA axis, and the stress response of the HPA axis. Any change in this interaction will strongly increase the possibility of neurodevelopmental disorders [55]. For neurogenesis, existing data indicate that maternal bacterial peptidoglycan can be transferred via the placenta and activate Toll-like receptor 2, altering the neural development of the embryo [73]. Furthermore, data suggest that early microbial colonization of the intestine regulates neurogenesis in the hippocampus [74].

With regard to CNS tissue macrophages, changes in gut microbiota of the host can significantly affect microglial homeostasis. Erny et al. show that limited gut microbiota composition in mice resulted in defective microglia and that bacterial fermentation products, such as short-chain fatty acids, regulate microglial properties [75]. With regard to the process of myelination, an Irish study elucidates the possible role of the host microbiota in regulating prefrontal cortex myelination in mice. Germ-free mice had a greater number of hypermyelinated axons, accompanied by the marked upregulation of genes related to myelination and myelin plasticity [76]. Finally, metabolites of gut microbiota seem to be part of the mechanisms regulating the integrity of the blood-brain barrier of the host, and can also affect paracellular permeability [77]. Animal model studies on mice provide evidence that metabolites, such as acetate, propionate, and sodium butyrate, are associated with adjusting the integrity of the blood-brain barrier [78,79].

Whether changes to gut microbiota are a cause of disease (or disorder), a result of disease (or disorder), or perhaps both, is a question that remains to be answered. This bidirectional axis of communication with various messenger pathways provides a wide field for further study [80].

## 5. Interactions between Gut Microbiota and Neurodevelopmental Disorders

### 5.1. Composition of Gut Microbiota in Different NDDs

Since the gut-brain axis was identified, it is evident that a causal association may exist between brain function or dysfunction and the host’s intestinal microorganisms. In a meta-analysis of 18 studies conducted by Iglesias-Vasquez et al., gut microbiota were compared in children with and without ASD. The microbiota mostly consisted of genera such as *Bacteroides*, *Parabacteroides*, and *Clostridium*, and these were significantly higher in children with ASD. However, children with ASD had smaller colonies of *Bifidobacterium*. This dysbiosis may play a role in the manifestation of ASD [8]. In a systematic literature review, Bundgaard-Nielsen et al. investigated the intestinal microbiota profile of individuals with ASD. Numerous studies identify an overall variation in β-diversity, even though no regular microbial variation between all studies has been reported [57]. A study from the United Kingdom reports that certain species, such as *Clostridium*, *Sutterella,* and *Lactobacillus*, are present in abundance in autistic children [81]. In line with these results, another study compared the composition of intestinal microbiota between individuals with ASD and healthy controls. *Lactobacillus*, *Bacteroides*, and *Clostridium* were consistently higher in people with ASD, while *Bifidobacterium*, *Blautia*, and *Prevotella*, among others, were constantly lower in the same population [82].

With regard to ADHD, a Danish review about the gut microbiome reports that the results were highly heterogeneous in patients with ADHD [57]. In contrast, in a review by Wang et al., no significant differences in a-diversity were observed in people with ADHD compared with healthy controls [56]. Table 1 summarizes data from studies on gut microbiome composition and their metabolites in children and young adults diagnosed with ASD or ADHD [72,83,84,85,86,87,88,89,90,91,92].

The disorders mostly studied were ASD and ADHD. The difference between sampling and microbial procedures remains a considerable challenge in the investigation of gut dysbiosis and its potential role in NDDs, not only in ASD and ADHD, but in other disorders as well.

### 5.2. Different Interventions and Possible Treatment Approaches

The clinical characteristics and evolution of NDDs may possibly be controlled by interventions targeting the gut microbiota such as the administration of antibiotics, probiotics, prebiotics, or even fecal microbiota transplantation (FMT) [93]. FMT (i.e., the administration of fecal material from a donor into the intestinal system of a recipient) remains a fairly new treatment; it needs to be further studied since patients could be at risk of serious adverse or side effects caused by the donor’s microbiome [94,95].

### 5.3. Diet and Regulation of NDDs

Elimination diets have been proposed as a possible means of regulating the symptoms and severity of NDDs. Several systematic reviews have studied the efficacy of gluten-free/casein-free diets in young patients with ASD. Data from high quality RCTs is scarce at present [96]. Elimination diets and their impact on children with ADHD were investigated in a few studies, but the data are of low quality [97,98]. Further research is imperative in order to reach safe conclusions. Although highly popular among patients with epileptic syndromes who are non-responsive to anticonvulsant medications, ketogenic diets appear controversial in the treatment of ASD. There is insufficient evidence to corroborate the effectiveness of such diets on ASD. A randomized clinical trial in Turku, Finland, reported that probiotic supplementation early in life might reduce the risk of ADHD later in childhood [90]. However, these data need to be supported by more robust evidence.

### 5.4. Probiotics and NDDs

Probiotics are live nonpathogenic bacteria and yeast administered to enhance and promote microbial balance, especially in the gastrointestinal system. They mostly consist of *Saccharomyces boulardii* yeast or the *Lactobacillus* and *Bifidobacterium* species [99]. Their function is to lower the intestinal pH, decrease colonization by pathogenic microbes, and regulate the host’s immune response [99]. Sivamaruthy et al. suggest that treatment with probiotics could be considered a complementary and alternative therapeutic supplement for ASD [100]. In an Egyptian prospective open-label study, 30 autistic children received probiotic supplementation for 3 months. After supplementation, the stool PCR of the patients showed increased colony numbers of *Lactobacilli* and *Bifidobacteria*, and significant improvement in the severity of autism symptoms was noticed [101].

### 5.5. Fecal Microbiota Transplantation and NDDs

Gastrointestinal symptoms, such as bloating and abdominal pain, are frequently described in people diagnosed with ASD [102]. ASD patients have a different intestinal microbiota composition to healthy controls [84,103]. Kang et al. conducted an open-label clinical trial on 18 young patients with ASD receiving daily FMT for almost 2 months. They reported that both gastrointestinal and behavioral ASD symptoms improved, and that this improvement persisted up to 24 months after discontinuation of treatment [104]. However, no placebo group was included in the study. In a case series conducted by Ward et al. [105], intestinal symptoms often improved after FMT and regression was noticed after antibiotics.

Data on gut microbiota in patients with Tourette Syndrome is quite scarce. The existing evidence on a possible association between the syndrome and intestinal microorganisms is mainly derived from rat model studies [106]. With regard to pediatric acute-onset neuropsychiatric syndrome (PANS) and pediatric autoimmune neuropsychiatric disorders associated with streptococcal infections (PANDAS) [107], Ding et al. report a transient decrease of tic severity in males with Tourette Syndrome after FMT [108]. FMT is primarily studied in animal models in relation to other neurological disorders; therefore, it requires further investigation to avoid any serious adverse or side effects [93,95].

### 5.6. Possible Role of the Gut Microbiome-Brain Axis in NNDs

Whether the gut-brain axis plays an important role in the manifestation of certain NDDs is not fully understood. The gut-brain axis is an interesting issue, and research supports the existence of multiple feedback loops between the gut and the nervous system. Interactions have been identified in the models of several neurological, psychiatric, and digestive disorders. A more in-depth understanding of the causative impact is certainly needed to clarify these findings in randomized control trials and to apply this knowledge to disease treatment [69,109]. Studies have attempted to further investigate this potential link. With regard to ADHD, diet has a significant role in the symptomatology [110]. A small case-control study proposed a potential mechanism. People with ADHD have more abundant *Bifidobacterium* in their gut microbiota compared with healthy controls, which could lead to the distinctive regulation of dopamine precursors located in the human intestine [111].

ASD is another NDD that has been studied to identify potential pathophysiological mechanisms. Genetic studies in mice provide evidence that a knockout of the shank 3 gene (autism candidate) results in intense autistic-like behavior in the tested animals [112]. Another study on BTRB mice (inbred mouse strain that mostly works as a preclinical model of autism symptom domains) reported that the mice showed prolonged intestinal motility with constipation. Furthermore, a breakdown in the gut’s permeability deficit was reported [113,114,115]. In addition to these 2 suggested genetic models, in utero exposure to other agents has also been studied. Maternal intake of valproic acid led to intestinal inflammation and dysregulation of gut microbiome in animal models [116].

The role of gut microbiota and its possible connection with the pathophysiology of NDDs such as ASD and ADHD needs to be further investigated (Figure 2) [69].

### 5.7. Possible Role of the Gut Virome/Mycome-Brain Axis in NNDs

Virome populations can affect human hosts in numerous ways. Eukaryotic viruses that infect human cells can cause infections, occasionally causing disease. Phages can affect the host both indirectly, by regulating bacterial composition and bacterial function, or directly, by triggering an immune response. Research on the interaction between the virome and human diseases is in the early stages. The virome has been thought to be a possible cause of autoimmune diseases [50]. In one study, changes in viral populations were both directly and inversely linked with the development of pediatric type 1 diabetes [117]. In several other studies, virome signatures were associated with pediatric and adult inflammatory bowel disease [118,119,120]. The gut virome remains an increasingly identified and recognized factor contributing to health and homeostasis of the human CNS. Altered host virome-brain communication is being implicated in neurodevelopmental and neurological disorders, such as ASD and Parkinson’s disease [121,122,123]. The vast field of the human virome is only beginning to be explored in various studies. The role of the gut mycome is still unclear [54], and this could provide an interesting field of future research, since relevant studies on this topic do not exist.

## 6. Conclusions

The pathophysiology of the complex spectrum of neurodevelopmental disorders has not yet been fully understood. The developing brain of the fetus, and subsequently, during the first years of infancy and early childhood, is highly vulnerable to environmental factors. Stressful events occurring during this time permanently alter brain structure and function, thereby increasing susceptibility to neurodevelopmental disorders [124]. In the brain, neuroinflammation presents as microglial activation and as elevated levels of pro-inflammatory cytokines [125,126]. The gut-brain axis involves bidirectional communication pathways between the gut and the brain, consisting of endocrine, neural, and inflammatory mechanisms. Brain and gut microbiome interactions are developed during the first years of childhood, but can be modified by factors such as diet, medication, and stress throughout life [127]. Several studies highlight the important role of the gut microbiome in the homeostatic health-disease balance [128,129,130]. As scientific knowledge advances, the establishment and function of gut microbiota seems more complicated.

Most of the existing knowledge on gut microbiota function was acquired over the last decade; thus, efforts should be made to obtain scientific data for a better and more thorough understanding of the gut-brain axis and its involvement in the development of NDDs. Areas of future research, among others, include the role of the gut microbiome and whether its composition can affect the pathophysiology, or even the therapeutic interventions in disorders, such as ASD or ADHD. Given that the symptomatology of NDDs is quite broad, a more holistic treatment approach should be considered [59]. Such treatment interventions consist of modulating the human gut microbiota, which could provide satisfactory alleviation of clinical symptoms, and significant improvement in the quality of life of patients with NDDs.

Furthermore, it could be researched whether children with a healthy gut microbiota composition are less likely to be diagnosed with NDDs. The possible interaction of gut virome and mycome could also provide a promising area for future research.

## Figures and Tables

**Figure 1 microorganisms-11-00487-f001:**
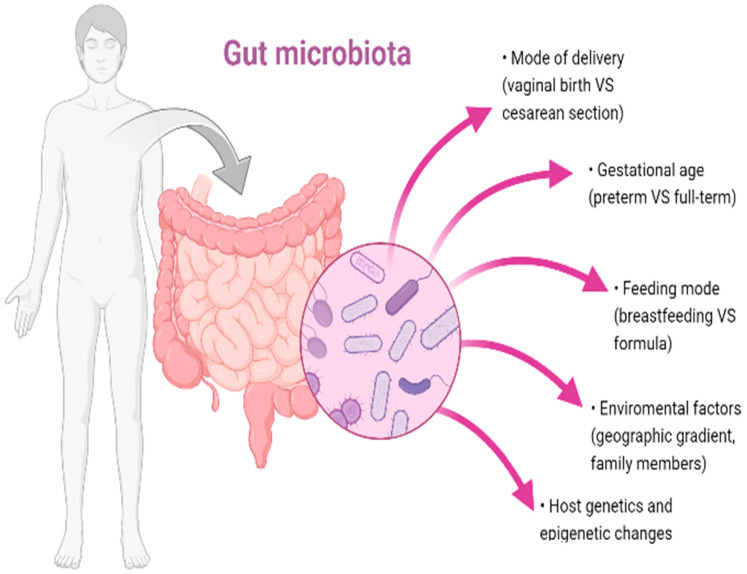
Factors that affect gut microbiota establishment.

**Figure 2 microorganisms-11-00487-f002:**
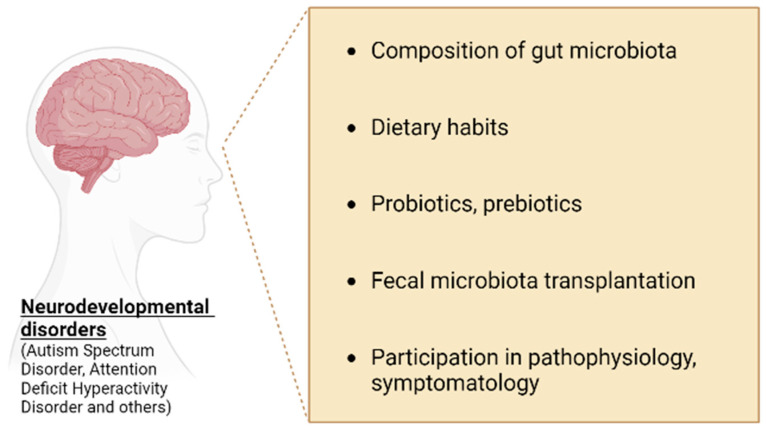
Interactions between gut microbiota and neurodevelopmental disorders.

**Table 1 microorganisms-11-00487-t001:** Bacterial composition differences in gut microbiota of people with NDDs (ASD, ADHD) compared with healthy controls, and their respective metabolite changes (when applicable) (NDDs: neurodevelopmental disorders, ASD: autism spectrum disorder, ADHD: attention deficit hyperactivity disorder, SCFA: short-chain fatty acid, FAA: free amino acids N/A: not applicable, ↑: increased bacterial population OR increased level of metabolites, when compared to the ones of healthy controls, ↓: decreased bacterial population OR decreased level of metabolites, when compared to the ones of healthy controls).

Study	Country, Year	Populations (N)	NDD	Gut Composition	Compare With Healthy Controls (Gut Composition)	Metabolites	Compare With Healthy Controls (Metabolites)
Finegold et al. [86].	USA, 2010	33 autistic children	ASD	*Bacteroidetes*	↑	N/A	N/A
				*Firmicutes*	↓		
				*Desulfovibrio*	↑		
				*Clostridium*	↑		
Adams et al. [83].	USA, 2011	58 autistic children	ASD	*Bifidobacter*	↓	SCFA (acetate, proprionate, and valerate)	↓
				*Lactobacillus*	↑		
Tomova et al. [88].	Slovakia, 2015	10 autistic children	ASD	*Lactobacillus*	↑	N/A	N/A
				*Desulfovibrio*	↑		
Finegold et al. [85].	USA, 2002	13 autistic children	ASD	*Clostridium*	↑	N/A	N/A
Parracho et al. [87].	UK, 2005	58 autistic children	ASD	*Clostridium*	↑	N/A	N/A
De Angelis et al. [84].	Italy, 2015	10 autistic children	ASD	*Bifidobacter*	↓	SCFA	↓
				*Clostridium*	↑	FAA	↓
Wang et al. [89].	Australia, 2012	23 autistic children	ASD	N/A	N/A	SCFA (fecal acetic, butyric, isobutyric, valeric, isovaleric)	↑
						Fecal ammonia	↑
						Calproic acid	No statistically important difference
Pärrty et al. [90].	Finland, 2015	6 autistic children	ADHD/ASD	*Bifidobacterium*	↓	N/A	N/A
Prehn-Kristensen et al. [91].	Germany, 2018	14 male ADHD patients	ADHD	*Prevotella*	↓	N/A	N/A
				*Parabacteroides*	↓		
				*Neisseria*	↑		
Jiang et al. [92].	China, 2018	51 children with ADHD	ADHD	*Faecalibacterium*	↓	N/A	N/A
Wang et al. [72].	Taiwan, 2020	30 children with ADHD	ADHD	*Fusobacterium*	↑	N/A	N/A

## Data Availability

Not applicable.

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
