# Peer review of "Gut Microbiome and Neurodevelopmental Disorders: A Link Yet to Be Disclosed"

_microorganisms, 2023, doi:10.3390/microorganisms11020487_

Round 1

Reviewer 1 Report

This manuscript deals with an exciting topic about the gut microbiome and neuro developmental disorders, explaining clearly the system but failing in their actual contribution to the field because the authors did not propose any new; please revise my detailed comments and suggestions in the attached file.

Author Response

Response to Reviewer 1 Comments

Point 1 (and the rest remarks relative to flora): - microbiome, flora is not accepted for microbial ecologist, indeed

Response 1: We thank you for your comment. The term flora has been successfully changed to microbiota or microbiome, depending on the context.

Point 2: - what about gut microbiome metabolic functions?

Response 2: We thank the reviewer for the remark. We have added the metabolic function of the gut microbiome as well.

Point 3: which ones?, please add some examples

Response 3: We thank you for your comment. We have added examples of disorders caused by this imbalance.

Point 4: - how we can link this concept with the concept of the holobiont?

Response 4: We thank the reviewer for this observation. We have made appropriate alterations throughout our text regarding the concept of the holobiont.

Point 5: actually we have some reviews related to this topic..

https://scholar.google.com.mx/scholar?as_ylo=2019&q=Neurodevelopmental+Disordders+and+gut+microbiome+review&hl=en&as_sdt=0,5

How the authors justify this new review article?

Response 5: We thank you for your observation. The text of the abstract was appropriately revised, so that the aim of the review is “to attempt to collect current knowledge on how gut health is linked with these disorders”.

Point 6: - in terms of composition, metabolism or what?

Response 6: Thank you for your remark. We have updated this part of the abstract, adding “in terms of composition and proper function”.

Point 7 (and other relative remarks about references): - needs a reference

Response 7: Thank you for the observation. References have been added throughout the text, when needed.

Point 8: what about scopus or WebOS

Response 8: We thank the reviewer for the comment. We have added to our search these online database platforms that will surely expand the knowledge to scientific subjects.

Point 9: - the authors revise all the narrative reviews about this same topic?

Response 9: Thank you for your question. The authors have collected and revised all scientific papers that could contribute information to the object of this review, which is to present collectively the existing knowledge on this topic. This part has been shortened for clarification reasons, since we conducted a narrative review.

Point 10: this phrase seems from a textbook of microbiology, i think that we can delete it

Response 10: We thank you for your remark. This part has been deleted to the revised text.

Point 11 (and other related comments): must be in italics

Response 11: Thank you for the comment. All terms that should be in italics have been appropriately changed.

Point 12: - please improve this section basen on this article

https://doi.org/10.3109/1040841X.2014.962478

Response 12: We thank the reviewer for this remark. We have added the relevant details in the text.

Point 13: - thus what is the conclusion of this argumentation, i think that just introduces more controversy about this unsolved topic...

Response 13: Thank you for your remark. In order to avoid any further controversy, this part has been deleted from the text.

Point 14: how the authors can assumed that those factors are "the most import" please clarify...

Response 14: We thank the reviewer for this comment. Since the gut microbiome gets mainly established during the first days of life of a human being, factors that play a role in the period of time were further expanded. The phrasing ''most important'' was altered and was better clarified for that purpose.

Point 15: the last two sections are very short, authors need to improve them or delete them,,

Response 15: Thank you for your observation. We deleted these two parts, in order to effectively increase the integrity of the chapter concerning factors that influence gut microbiota.

Point 16: ok this section was so interesting, please improve it

Response 16: Thank you for your comment. Further details regarding geographic gradient and gut microbiome, and additional information about Bifidobacteria concertration among siblings were included in the text. Extra data about the last cited study were added as well.

Point 17 (and related comments): which types of studies need to develop to understand this complex relationship...,

Response 17: We thank you for this question. Prospective studies with high-risk populations, such as preterm infants, babies of diabetic mothers and so on, could possibly offer a chance to study their respective gut microbiome.

Point 18 (and relative comments about figures): I think that this figure must be at the begining of the last paragraphs as an introductory figure...

Response 18: Thank you for this observation. The figures have been successfully modified for clarification reasons and transfered to a different position amongst text.

Point 19: section 2 actually is disconnected from section 3, please improve it with a more organic "transition"

Response 19: We thank you for this remark. We have included an introductory paragraph that associates gut microbiome and neurodevelopmental disorders.

Point 20: how can ligate this term with gut microbiome dysbiosis of individuals diagnosed with NDDs??

Response 20: We thank the reviewer for this observation. NDDs are a wide group of diseases that affect many aspects of life of a human being. They have multifactorial causation and various symptomatology, thus having an impact on the quality of life of a person. By including this paragraph, we highlight the importance of studying all possible factors that could affect NDDs, such as the gut microbiome.

Point 21: just probiotics?, I think that the solution of this isssue depends on a more holistic solution, what the authors think about this assumption? please clarify

Response 21: We thank the reviewer for this comment. More information about the holistic treatment approach for NDDs has been added.

Point 22: this refers to metadata from studies.. how we can exploit all the information presented in the metadata to improve our analysis??

Response 22: Thank you for your comment. In order to provide information about the topic that is of higher quality and better understood, this part has been deleted by the authors.

Point 23: how the authors think about the recent warns of FDA related to FMT...

https://www.fda.gov/safety/medical-product-safety-information/fecal-microbiota-transplantation-safety-alert-risk-serious-adverse-events-likely-due-transmission

Response 23: We thank the reviewer for this comment. We have included further clarification in the text, stating clearly that FMT is a fairly new treatment approach in the medical community, that also includes risks of serious adverse effects.

Point 24: the authors revise these patents???..

US11357801B2 United States

US20220088089A1 United States

and these articles?

https://www.medrxiv.org/content/10.1101/2022.10.26.22281525v1

Response 24: We thank you for your comment. We revise this open-label study https://pubmed.ncbi.nlm.nih.gov/28686541/, which studies the population of 30 autistic children, after the supplementation of probiotics for 3 months. The suggested article provides additional, interesting and informative data about given synbiotics to both children and adults.

Point 25: accordinh their experience and knowledge, what the authors think about this sentence...

Response 25: We thank the reviewer for this comment. The original text has been modified, so that the context about gut-brain axis is properly presented to the readers. The gut-brain axis is an extremely interesting scientific subject, and research has shown with multiple feedback loops. Interactions have been identified in models of several neurological, psychiatric and digestive disorders. A more comprehensive understanding of causative influences and randomized control trials are needed to identify these findings and apply this knowledge to diseases treatment.

Point 26: actually this is not new, because we have a lot of data related to these interactions, therefore, what was the main input of this review article

Response 26: Thank you for your observation. So far the scientific data for this subject has been increasing, and the aim of this review is to collect the current knowledge under a specific spectrum.

Point 27: Authors need to imprive this section, because, actually did not contribute with the advances of the field and did not explain any potential strategy to improve the knowledge o gut brain axis...

Response 27: Thank you for this observation. This chapter of the review has been revised and improved with additional sentences. The aim of this review is to collect and present relative data that showcase the interaction of gut microbiome and NDDs and also to propose possible ideas for future research.

Reviewer 2 Report

In this narrative review, Iliodromiti et al, describe the current state of knowledge about the connection between the gut microbiome and neurodevelopmental disorders (NDDs). The choice of topic is actual and relevant, however, multiple issues need to be addressed before publication.

- The manuscript needs at least one more round of English editing. There are typos and grammatical errors scattered throughout the text.

- The quality and complexity of figures need to be improved significantly. Please colorize them and instead of creating basically simple flow charts, illustrate these figures more richly. This will attract more eyes to read the article.

- In the last paragraph of the Introduction section, the Authors detail the methods and sources of literature search. Since this is not a systematic review, this kind of description is not really needed. 

- It would be a great addition (and would also boost the number of references) if the Authors dedicated a chapter to studies that found or hypothesized any kind of connection between the gut mycobiome/virome and NDDs. Also, it would be nice to add a paragraph about these microorganisms in the first part of the manuscript where the establishment of the microbiome in newborns and infants is detailed.

- Every narrative review needs a comprehensive and easy-to-read Table for its most important findings. Please include a table about identified microbial taxa and metabolites associated with NDDs.

Author Response

Response to Reviewer 2 Comments

Point 1: - The manuscript needs at least one more round of English editing. There are typos and grammatical errors scattered throughout the text.

Response 1: We thank you for your comment. Some modifications have been made in our text.

Point 2: - The quality and complexity of figures need to be improved significantly. Please colorize them and instead of creating basically simple flow charts, illustrate these figures more richly. This will attract more eyes to read the article.

Response 2: We thank the reviewer for the remark. The figures have been according modified. They have been colorized and enriched with illustrated details.

Point 3: - In the last paragraph of the Introduction section, the Authors detail the methods and sources of literature search. Since this is not a systematic review, this kind of description is not really needed.

Response 3: We thank you for your comment. We have summarized this paragraph in our text, for clarification reasons. A less extended description of the methodology has replaced the initial sentences.

Point 4: - It would be a great addition (and would also boost the number of references) if the Authors dedicated a chapter to studies that found or hypothesized any kind of connection between the gut mycobiome/virome and NDDs. Also, it would be nice to add a paragraph about these microorganisms in the first part of the manuscript where the establishment of the microbiome in newborns and infants is detailed.

Response 4: We thank the reviewer for this most interesting observation. The assessment of a possible interaction between neurodevelopmental disorders and virome/mycome provides a non explored study area to the researchers. The data provided by the scientific papers were taken down and appropriate information about them was added in our text.

Point 5: - Every narrative review needs a comprehensive and easy-to-read Table for its most important findings. Please include a table about identified microbial taxa and metabolites associated with NDDs.

Response 5: We thank you for your observation. We have included a table with summarized data for microbial taxa associated with NDDs, when applicable, in order to assist the readers with the most important findings.

Round 2

Reviewer 1 Report

The authors improved their manuscript clearly; I think it is ready for publication, with minor changes in spelling.

Author Response

Dear reviewer your recommendations and suggestions were helpful, and the initial manuscript was corrected according to them and we realized that you have been satisfied with the revised manuscript.

You have expressed your concern about one more round of English spelling, and we were worked on editing this.

Thank you for guidance and mindful comments.

Kind regards,

Zoi Iliodromiti

Reviewer 2 Report

The Authors sufficiently addressed all the Reviewer's remarks regarding the content of the manuscript, and the quality of the Figures has improved as well. 

My only concern is still the English. Despite the fact that grammatical mistakes have been corrected, the phrasing and the smoothness of English language usage still have some issues.

I would recommend one more round of English editing, especially for the newly incorporated parts.  

Author Response

(The authors gave the same response as above.)
